# JAK2 Inhibitors and Emerging Therapies in Graft-Versus-Host Disease: Current Perspectives and Future Directions

**DOI:** 10.3390/biomedicines13071527

**Published:** 2025-06-23

**Authors:** Behzad Amoozgar, Ayrton Bangolo, Abdifitah Mohamed, Charlene Mansour, Daniel Elias, Christina Cho, Siddhartha Reddy

**Affiliations:** 1Department of Hematology and Oncology, John Theurer Cancer Center, Hackensack University Medical Center, Hackensack, NJ 07601, USA; ayrton.bangolo@hmhn.org; 2Department of Internal Medicine, University of Washington School of Medicine, Seattle, WA 98195, USA; drsharif@uw.edu; 3Department of Internal Medicine, Rutgers New Jersey Medical School, Newark, NJ 07103, USA; cm1621@njms.rutgers.edu (C.M.); dannyelias28@gmail.com (D.E.); 4Division of Stem Cell Transplant and Cellular Therapy, John Theurer Cancer Center, Hackensack University Medical Center, Hackensack, NJ 07601, USA; christina.cho@hmhn.org (C.C.); siddhartha.reddy@hmhn.org (S.R.)

**Keywords:** GVHD, JAK2 inhibitors, ruxolitinib, axatilimab, mesenchymal stromal cells

## Abstract

Graft-versus-host disease (GVHD) remains a significant barrier to the success of allogeneic hematopoietic stem cell transplantation (allo-HSCT), contributing to long-term morbidity and non-relapse mortality in both pediatric and adult populations. Central to GVHD pathophysiology is the Janus kinase (JAK)-signal transducer and activator of transcription (STAT) pathway, where JAK2 mediates key pro-inflammatory cytokines, including IL-6, IFN-γ, and GM-CSF. These cytokines promote donor T cell activation, effector differentiation, and target organ damage. The introduction of ruxolitinib, a selective JAK1/2 inhibitor, has transformed the treatment landscape for steroid-refractory acute and chronic GVHD, leading to improved response rates and durable symptom control. However, its limitations—such as cytopenias, infectious complications, and incomplete responses—have catalyzed the development of next-generation agents. In 2024, the FDA approved axatilimab, a CSF-1R inhibitor that targets monocyte-derived macrophages in fibrotic chronic GVHD, and remestemcel-L, an allogeneic mesenchymal stromal cell therapy, for pediatric steroid-refractory acute GVHD. Both agents offer mechanistically distinct and clinically meaningful additions to the therapeutic armamentarium. In parallel, emerging combination strategies involving JAK2 inhibitors and novel biologics show promise in enhancing immune tolerance while preserving graft-versus-leukemia (GvL) effects. Recent advances in biomarker development, such as the MAGIC Algorithm Probability (MAP), are enabling early risk stratification and response prediction. The integration of these tools with organ-specific and personalized approaches marks a shift toward more precise, durable, and tolerable GVHD therapy. This review highlights the current state and future direction of JAK2 inhibition and complementary therapies in the evolving GVHD treatment paradigm.

## 1. Introduction

Graft-versus-host disease (GVHD) remains a major cause of morbidity and non-relapse mortality following allogeneic hematopoietic stem cell transplantation (allo-HSCT), representing a significant obstacle to its curative potential in both malignant and non-malignant hematologic diseases. GVHD results from an alloimmune attack in which donor T lymphocytes recognize host tissues as foreign. Clinically, it manifests in two distinct but sometimes overlapping phenotypes: acute GVHD (aGVHD), which predominantly affects the skin, liver, and gastrointestinal tract; and chronic GVHD (cGVHD), which presents later with systemic fibrotic and autoimmune-like features involving the eyes, lungs, oral mucosa, fascia, and other organs [1].

While GVHD has historically been treated with systemic corticosteroids and broad immunosuppression, outcomes remain suboptimal, especially in steroid-refractory disease. Advances in molecular immunology have identified the Janus kinase (JAK)–signal transducer and activator of transcription (STAT) signaling axis—particularly JAK2—as a pivotal therapeutic target. JAK2 mediates intracellular signaling of multiple pro-inflammatory cytokines implicated in GVHD pathogenesis, including interferon-γ (IFN-γ), interleukin-6 (IL-6), granulocyte-macrophage colony-stimulating factor (GM-CSF), and interleukin-12 (IL-12) [2,3]. These cytokines promote expansion of alloreactive CD4^+^ and CD8^+^ T cells, enhance MHC class II expression on antigen-presenting cells (APCs), and impair regulatory T cell (Treg) recovery, contributing to sustained inflammation and tissue damage (Figure 1) [4,5,6].

Pharmacologic inhibition of JAK2 has been shown to suppress STAT1/STAT3-driven cytokine responses, reduce Th1/Th17 polarization, and preserve Treg stability, while maintaining graft-versus-leukemia (GvL) effects in preclinical models [7,8,9,10]. These findings provided the rationale for the clinical development of ruxolitinib, a selective JAK1/2 inhibitor that was subsequently approved by the FDA for steroid-refractory aGVHD in 2019 and cGVHD in 2021 [11,12,13].

However, limitations of ruxolitinib—including cytopenias, infectious complications, and incomplete responses—have catalyzed efforts to develop complementary and next-generation strategies. These include novel JAK2-targeted agents such as baricitinib and pacritinib, as well as recently approved non-T cell-targeted therapies such as axatilimab (a CSF-1R monoclonal antibody) and remestemcel-L (a mesenchymal stromal cell therapy) [5]. In parallel, biomarker-guided and combination-based approaches are emerging to refine treatment intensity and personalize care.

This review focuses on the biologic rationale and clinical application of JAK2 inhibition, explores the role of newly approved and investigational agents, and highlights the next frontiers in GVHD management through precision immunomodulation.

## 2. JAK2 Inhibitors in GVHD: Mechanisms and Clinical Applications

### 2.1. Mechanism of JAK2 Inhibition in GVHD

The JAK family includes four non-receptor tyrosine kinases—JAK1, JAK2, JAK3, and TYK2—that transduce cytokine-mediated signals through the JAK–STAT (signal transducer and activator of transcription) pathway. Upon ligand binding, cytokine receptors activate JAKs, which phosphorylate downstream STAT proteins. These then dimerize and translocate to the nucleus to regulate gene expression programs involved in immune cell activation, proliferation, and differentiation [6].

As mentioned previously, JAK2 plays a pivotal role in transducing signals from key pro-inflammatory cytokines implicated in GVHD pathogenesis, including IL-6, IFN-γ, GM-CSF, and IL-12 [7]. Genetic deletion of JAK2 in murine models leads to embryonic lethality due to hematopoietic failure, underscoring its essential role in immune and hematopoietic signaling [7].

In the post-transplant setting, JAK2 signaling promotes expansion of alloreactive CD4^+^ and CD8^+^ T cells, upregulates MHC class II expression on antigen-presenting cells (APCs), and enhances CXCR3-mediated chemotaxis, collectively sustaining GVHD propagation [8,9]. Pharmacologic inhibition of JAK2 suppresses STAT1/STAT3 signaling and reduces the production of IFN-γ, IL-6, and IL-12, while simultaneously promoting the expansion and phenotypic stability of regulatory T cells (Tregs), thereby restoring immune tolerance [10,11].

Preclinical models further demonstrate that JAK2 inhibition suppresses Th1 and Th17 polarization, mitigates inflammatory tissue damage, and preserves graft-versus-leukemia (GvL) activity [8,10]. These findings position JAK2 as a central immunologic node in GVHD pathogenesis and a rational therapeutic target for both aGVHD and cGVHD.

### 2.2. Clinical Trials and FDA Approvals

Ruxolitinib (Jakafi^®^) (Incyte Corporation, Wilmington, DE, USA) is a selective JAK1/2 inhibitor, was the first agent approved by the U.S. Food and Drug Administration (FDA) for the treatment of steroid-refractory acute GVHD (aGVHD) in 2019 and chronic GVHD (cGVHD) in 2021 [12]. Its approval was supported by the pivotal REACH2 and REACH3 trials.

In the phase III REACH2 study of steroid-refractory aGVHD, ruxolitinib significantly improved the overall response rate (ORR) at day 28 compared to the best available therapy (62% vs. 39%; *p* < 0.001), with durable responses and improved failure-free survival [7]. The REACH3 trial, conducted in patients with steroid-refractory cGVHD, similarly demonstrated superior ORR, improved symptom burden, and greater steroid-sparing effects versus the investigator’s choice [13]. These findings established ruxolitinib as a new standard of care in steroid-refractory GVHD. Real-world studies have corroborated its clinical benefit, though adverse events such as cytopenias and increased infection risk necessitate close monitoring [14].

#### Comparison with Other JAK Inhibitors

Alternative JAK inhibitors are under active investigation to optimize the therapeutic window and reduce toxicity. Baricitinib, a JAK1/2 inhibitor that spares JAK3, has demonstrated favorable effects on Treg preservation and less disruption of IL-2 signaling compared to ruxolitinib [7]. In murine GVHD models, baricitinib more effectively suppressed Th1 and Th17 responses, expanded Tregs, and improved survival while promoting intestinal epithelial regeneration through EGFR signaling [10,15].

Itacitinib, a JAK1-selective inhibitor, has shown reduced myelosuppression in early trials. However, its efficacy in aGVHD has been limited, likely due to insufficient JAK2 inhibition. In the phase III GRAVITAS-301 trial, itacitinib combined with corticosteroids led to a higher day-28 ORR versus placebo (74% vs. 66%), but this difference did not meet the prespecified threshold for statistical significance. No substantial improvements in failure-free survival or non-relapse mortality were observed [16].

Pacritinib, a dual JAK2/FLT3 inhibitor, has demonstrated promising preclinical activity by reducing STAT3^+^ CD4^+^ T cell expansion, attenuating tissue damage, and preserving GvL effects. In murine GVHD models, pacritinib suppressed IL-6 and GM-CSF production while sparing IL-2 signaling essential for Treg homeostasis. Although clinical data are in the early stage, these findings support its potential role in GVHD immunomodulation [17] [Table 1].

## 3. Recent FDA-Approved Therapies in GVHD: Expanding Treatment Options

### 3.1. Axatilimab (Niktimvo): CSF-1R Inhibition

Axatilimab is a humanized IgG4 monoclonal antibody targeting the colony-stimulating factor 1 receptor (CSF-1R), a key regulator of monocyte and macrophage differentiation. In chronic GVHD (cGVHD), pathogenic monocyte-derived macrophages contribute to tissue inflammation and fibrosis, particularly in the skin, gastrointestinal tract, and lungs [18]. By blocking CSF-1R signaling, axatilimab selectively depletes profibrotic macrophages while sparing hematopoietic progenitors, offering a targeted approach to immune modulation [19,20].

The AGAVE-201 study, a phase II multicenter trial, evaluated axatilimab monotherapy in patients with heavily pretreated, refractory or recurrent cGVHD. Across three dosing cohorts, axatilimab achieved overall response rates (ORRs) ranging from 50% to 74%, with organ-specific improvement in NIH metrics and reductions in symptom burden, including resolution of skin sclerosis [21]. Importantly, clinical responses were observed regardless of prior exposure to ruxolitinib, ibrutinib, or belumosudil [21].

Mechanistically, axatilimab targets a unique axis in chronic inflammation by depleting CSF-1-dependent donor macrophages, which persist long-term in affected organs and sustain profibrotic signaling via TGF-β and fibroblast activation [20,22]. In murine models, the CSF-1R blockade reduced tissue-resident macrophage populations and ameliorated cGVHD histopathology, further validating the therapeutic target [20].

In August 2024, axatilimab received FDA approval as a third-line treatment for cGVHD in adults and children ≥ 40 kg who had failed at least two prior lines of systemic therapy [22]. It was concurrently incorporated into the NCCN Guidelines for cGVHD management [21,23]. The drug is administered via intravenous infusion at 0.3 mg/kg every two weeks. Adverse effects are generally mild and manageable, including transient transaminase elevations and periorbital edema related to tissue-resident macrophage depletion [19,24].

### 3.2. Remestemcel-L (Ryoncil): Mesenchymal Stromal Cell Therapy

Remestemcel-L is an ex vivo expanded, allogeneic bone marrow-derived mesenchymal stromal cell (MSC) product approved by the FDA in December 2024 for the treatment of pediatric steroid-refractory acute GVHD (SR-aGVHD) [25]. MSCs exhibit potent immunomodulatory and tissue-repair properties, including expansion of regulatory T cells (Tregs), suppression of Th1/Th17 responses, and secretion of anti-inflammatory cytokines such as IL-10 and TGF-β [26,27].

The pivotal GVHD001 study, a single-arm phase III trial involving 54 pediatric patients with SR-aGVHD, demonstrated an ORR of 70.4% at day 28, significantly exceeding the prespecified historical control rate of 45% (*p* < 0.001). Responses were durable, with an ORR of 74.1% sustained through day 100. Moreover, patients who achieved response at day 28 had significantly improved day-100 survival compared to non-responders (86.8% vs. 47.1%, *p* < 0.0001) [28].

Although MSC-based therapies have been studied for nearly two decades, reproducibility, regulatory hurdles, and product variability have limited prior clinical adoption. Earlier-generation products, such as Prochymal, failed to meet efficacy endpoints in adult GVHD populations. However, remestemcel-L demonstrated efficacy particularly in younger patients with hepatic or gastrointestinal aGVHD, a benefit attributed in part to its standardized allogeneic donor source, consistent ex vivo expansion protocols, and improved product stability, which are features that overcome prior limitations in MSC therapy reproducibility [29,30].

Remestemcel-L’s approval represents the first MSC-based product authorized in the United States for any indication and reflects a broader shift toward tissue-protective, non-lymphodepleting therapies in GVHD. Ongoing research aims to optimize MSC source, dosing, and adjunctive regimens, potentially expanding their role in both acute and chronic GVHD.

## 4. Combination Strategies: JAK2 Inhibitors and Emerging Therapies

### 4.1. Synergistic Effects of JAK2 Inhibition with Immunomodulatory Agents

JAK2 inhibitors, particularly ruxolitinib and baricitinib, are increasingly being explored in combination strategies designed to augment immunosuppression, preserve immune reconstitution, and promote tissue repair. These agents suppress key pro-inflammatory cytokines, including interferon-gamma (IFN-γ) and interleukin-6 (IL-6), downregulate antigen-presenting cell (APC) activity, and expand regulatory T cells (Tregs), thereby promoting a tolerogenic immune environment in both acute and chronic GVHD [9,31].

Baricitinib, in particular, has demonstrated synergy with the mammalian target of rapamycin (mTOR) inhibitors in preclinical models. This combination enhances Treg recovery and significantly reduces GVHD severity compared to either agent alone [32]. Additionally, baricitinib has been shown to promote epithelial regeneration via epidermal growth factor receptor (EGFR) signaling, which supports intestinal barrier repair while attenuating alloimmune tissue injury [15]. These dual effects on immune modulation and tissue protection provide a strong rationale for combinatorial approaches in both prophylactic and therapeutic settings.

Given the central role of mucosal injury in GVHD pathogenesis—particularly in the gastrointestinal tract—combinations involving JAK inhibitors and agents that enhance epithelial integrity, such as IL-22 analogs or mesenchymal stromal cells (MSCs), may be especially valuable. Furthermore, integration with immunoregulatory agents that modulate antigen presentation (e.g., CD83 or CD40 blockade) could further dampen alloimmune activation. In the prophylactic setting, pairing JAK2 inhibition with established strategies like post-transplant cyclophosphamide (PTCy) or abatacept may enable steroid-sparing regimens that maintain graft-versus-leukemia (GvL) activity [33].

While clinical data on these combinations are limited, their mechanistic rationale is supported by extensive preclinical evidence. Ongoing and future trials will be critical to determine optimal dosing, sequencing, and safety profiles for these synergistic regimens.

### 4.2. Combinatorial Approaches with Axatilimab or Remestemcel-L

Combinatorial regimens targeting both adaptive and innate immune pathways are of increasing interest in GVHD, particularly for refractory or fibrotic cGVHD subtypes. Axatilimab, an anti-CSF-1R monoclonal antibody, depletes monocyte-derived macrophages involved in fibrotic tissue remodeling. Its mechanism complements JAK2 inhibitors, which primarily modulate T cell-driven inflammation [19]. While no prospective clinical trials have evaluated the co-administration of axatilimab with ruxolitinib or baricitinib, retrospective series and expert consensus suggest potential synergy, particularly in patients with cutaneous or sclerotic cGVHD who progressed on prior JAK inhibition [19,34].

Remestemcel-L, an allogeneic MSC product, exerts immunomodulatory and tissue-reparative effects through the expansion of Tregs, secretion of IL-10 and TGF-β, and suppression of APC maturation [27]. Although preclinical studies and in vitro models demonstrate that MSCs can enhance tolerogenic immune phenotypes and may act synergistically with JAK inhibition [35,36,37,38], direct clinical evidence of JAK–MSC combinations remains lacking. A related trial combining MSCs with basiliximab (IL-2R antagonist) showed improved ORR in pediatric SR-aGVHD, supporting the feasibility of MSC-based combinations [39].

### 4.3. Preserving Graft-vs.-Leukemia Effects in Combination Therapies

An essential goal of GVHD therapy is to maintain graft-versus-leukemia (GvL) effects while minimizing alloimmunity. JAK inhibitors, particularly ruxolitinib and baricitinib, have demonstrated this balance in murine models by selectively suppressing Th1/Th17 polarization while preserving cytotoxic CD8^+^ T cell responses [8]. Baricitinib, through sparing of JAK3–STAT5 signaling, promotes Treg expansion and reduces MHC II and co-stimulatory molecule expression on APCs, thus attenuating GVHD without eliminating GvL activity [8].

When used in combination, the impact on GvL is more complex and depends on the immune cell subsets targeted. Agents like axatilimab act predominantly on macrophage lineages and are less likely to impair cytotoxic T cell function. In contrast, MSCs may dampen T cell effector function more broadly, which could theoretically impair GvL if not properly sequenced or dosed.

Although no clinical studies have yet assessed GvL preservation in JAK-based combination regimens, preclinical models support the hypothesis that non-depleting agents (e.g., JAK inhibitors, MSCs, CSF-1R blockade) can modulate GVHD while retaining anti-leukemic immunity. Furthermore, combinations with PTCy or abatacept are under investigation for their ability to enhance immune tolerance without impairing tumor surveillance [1,40].

## 5. Safety, Adverse Effects, and Challenges

### 5.1. Comparative Analysis of Toxicities and Adverse Events

JAK2 inhibitors, while effective in steroid-refractory GVHD, are associated with well-documented toxicities, most notably cytopenias and infectious complications. In both the REACH2 and REACH3 trials, ruxolitinib was linked to grade 3–4 anemia and thrombocytopenia in up to 50% of patients [41,42,43]. These cytopenias frequently required dose reductions, growth factor support, or therapy interruption.

Other JAK inhibitors show variable hematologic toxicity. Itacitinib, due to its JAK1 selectivity, was associated with lower rates of myelosuppression, although gastrointestinal toxicity and infection risk persisted [16]. Baricitinib demonstrated favorable immune-modulatory effects in preclinical models but raised concerns about thrombotic risk and metabolic changes in autoimmune disease populations [44]. Pacritinib also exhibited cytopenias and potential QTc prolongation in early trials [17].

Beyond acute toxicities, longitudinal safety data remain limited for most agents. Reports from myelofibrosis and GVHD registries suggest potential long-term risks of secondary malignancies, including non-melanoma skin cancers and lymphoid neoplasms during extended ruxolitinib exposure [2,10]. Furthermore, drug–drug interactions, particularly through CYP3A4 metabolism (e.g., with azoles or calcineurin inhibitors), may exacerbate both cytopenias and infectious risk [1].

### 5.2. Management of Immune Dysregulation and Infection Risks

Immunosuppression with JAK2 inhibitors suppresses alloimmunity but can impair pathogen surveillance, increasing risk for opportunistic infections. Reactivation of CMV, EBV, BK virus, and invasive fungal infections (IFIs) is well documented, particularly in the context of prolonged immunosuppressive exposure [41,43].

Letermovir prophylaxis and weekly viral load monitoring are increasingly recommended in patients on JAK inhibitors, especially those with prior viral seropositivity. These preventive strategies have been shown to reduce CMV-related morbidity and support the continued use of JAK inhibitors in high-risk populations [43]. Notably, axatilimab and remestemcel-L have not demonstrated similar infectious risk profiles, likely due to cell- or lineage-specific targeting [19,28,29].

Nonetheless, patients with cGVHD often receive multi-agent immunosuppression, so multifaceted infection prophylaxis (antivirals, antifungals, and pneumocystis prophylaxis) remains a cornerstone of supportive care [45].

### 5.3. Considerations for Long-Term Therapy in Chronic GVHD

The management of chronic GVHD often requires extended therapy, and cumulative toxicity becomes a concern. Long-term ruxolitinib use has been linked to progressive cytopenias, weight gain, dyslipidemia, and secondary malignancies in registry data [2,10]. Further, drug–drug interactions via CYP3A4 metabolism, particularly with azoles and calcineurin inhibitors, may exacerbate these risks [1].

Agents such as axatilimab and belumosudil, which target profibrotic pathways, may be more suitable for fibrotic-predominant cGVHD. However, long-term safety data remain limited, and there are no validated sequencing algorithms or biomarker-driven personalization strategies in current practice [19,21,46].

Future efforts should focus on tailoring therapy duration, immunologic monitoring, and defining longitudinal toxicity profiles, particularly in younger transplant survivors [19,21,46]. Personalized approaches that incorporate biomarkers of immune reconstitution and patient-specific risk factors may optimize treatment efficacy while minimizing adverse effects [47]. Additionally, long-term surveillance for late complications such as secondary malignancies and organ dysfunction is critical to improving overall survivorship outcomes in this population [48].

## 6. Future Perspectives and Unmet Needs

### 6.1. Ongoing Trials Evaluating Next-Generation JAK Inhibitors and Novel Biologics

The therapeutic landscape of GVHD continues to evolve with the investigation of next-generation JAK inhibitors and novel immunomodulatory agents that offer mechanistic diversity and the potential for improved safety profiles. Selective agents such as baricitinib, itacitinib, and pacritinib are currently being explored in both prophylactic and treatment settings, either as monotherapies or in combination with established platforms such as post-transplant cyclophosphamide (PTCy) and abatacept [1]. These agents differ in their selectivity for JAK isoforms and associated toxicity profiles, providing an opportunity for personalized immunosuppression tailored to individual disease phenotypes and tolerability thresholds [49]. For example, emerging data suggest that JAK1-selective inhibitors such as itacitinib may reduce the incidence of cytopenias and opportunistic infections, making them attractive for frontline or less aggressive GVHD presentations. In contrast, broader inhibitors like ruxolitinib and pacritinib may retain greater efficacy in steroid-refractory cases, though they carry higher risk for hematologic adverse events [50].

Beyond kinase inhibition, multiple biologics are being developed to target tissue-specific repair pathways and non-lymphoid components of GVHD. One such class includes IL-22 analogs, such as the recombinant fusion protein F-652, which is being studied for its capacity to promote epithelial regeneration in gastrointestinal GVHD. In a Phase II trial, F-652 administered in combination with corticosteroids yielded a 70% overall response rate by day 28, underscoring its potential to restore mucosal integrity and reduce GI-specific damage without broadly suppressing immunity [51].

In addition to cytokine mimetics, several trafficking-modulatory strategies are under investigation. Sphingosine-1-phosphate (S1P) receptor modulators, such as ozanimod, aim to prevent effector T cell egress from lymphoid organs and limit infiltration into target tissues, thereby mitigating alloimmune injury while sparing regulatory and memory compartments. Preclinical studies and early-phase trials in inflammatory disorders support their tolerability and immunomodulatory potential, but GVHD-specific data are limited and require further validation [52,53].

Another promising category includes bromodomain and extraterminal domain (BET) inhibitors, which regulate epigenetic transcriptional programs involved in T cell exhaustion, dendritic cell activation, and inflammatory cytokine production. In preclinical GVHD models, BET inhibition has been shown to decrease Th17 polarization and fibrotic progression, with ongoing trials evaluating these agents in hematologic malignancies and chronic inflammation [54,55]. Epigenetic modifiers may also synergize with JAK inhibitors by modulating overlapping transcriptional networks that drive GVHD persistence.

As these novel agents progress through clinical development, integration into combinatorial regimens and biomarker-guided algorithms will be essential to optimize efficacy while minimizing overlapping toxicities.

### 6.2. Development of Biomarkers for Predicting Treatment Response

Biomarker-guided therapy has become a cornerstone of precision medicine in GVHD, with the MAGIC Algorithm Probability (MAP), integrating serum ST2 and REG3α, representing the most validated composite model to date. MAP scoring has been incorporated into prospective studies for risk stratification, treatment intensification, and dynamic monitoring, demonstrating superior prognostic accuracy over clinical grading systems alone [56,57,58,59,60,61]. However, relying solely on this two-marker system may limit the ability to capture the full spectrum of GVHD biology, particularly in chronic GVHD (cGVHD) and in non-gastrointestinal target organs.

Expanding the biomarker repertoire is therefore critical. Several candidate markers have emerged, including CXCL9 and CXCL10, which are elevated in IFN-γ-driven inflammation and correlate with skin and musculoskeletal cGVHD activity [62]. BAFF (B-cell activating factor) has been linked to B-cell dysregulation, particularly in sclerotic and ocular manifestations of cGVHD [41,62]. Additional markers such as MMP-3, soluble CD163, osteopontin, and elafin reflect fibrotic and epithelial damage across target organs and are under active investigation in both adult and pediatric cohorts [41,60,62,63]. Despite these advances, none of these biomarkers are validated for routine clinical decision-making, and their incorporation into standard care remains investigational.

To advance these biomarker candidates toward clinical adoption, several concrete steps must be taken. First, assay standardization—including calibration of ELISA platforms, inter-laboratory reproducibility, and biologically meaningful cutoffs—must be achieved to allow for cross-study comparisons [64]. Second, these biomarkers should be prospectively incorporated into multicenter clinical trials as embedded stratification tools rather than post hoc exploratory endpoints. For instance, trials could stratify patients based on baseline CXCL9 or BAFF levels to compare immunosuppressive strategies or early therapeutic escalation. Third, adaptive trial designs should be implemented, in which dynamic biomarker changes (e.g., MAP decline by day 7–14) trigger predefined clinical decisions. This approach has already been piloted in MAP-driven trials of natalizumab and ruxolitinib [60,65]. Finally, biomarker-guided care must be evaluated for clinical utility, turnaround time, and cost-effectiveness—metrics essential for future reimbursement and widespread adoption.

Several research consortia, including the NIH Consensus Project and the Children’s Oncology Group GVHD Biomarker Network, are currently working to harmonize these platforms and validate multianalyte classifiers [60,61,66]. The ultimate goal is a precision GVHD management model in which early treatment is guided not only by clinical phenotype but also by immune and tissue biomarkers that predict severity, response, and relapse risk.

### 6.3. Personalized Approaches for GVHD Management

The complexity and heterogeneity of GVHD necessitate a personalized medicine framework. The increasing importance of tailoring GVHD therapies is underscored by risk-stratified algorithms, which have utilized clinical and biomarker criteria to identify aGVHD patients. As a result, lower-risk cases are able to benefit from a treatment regimen that spares the use of corticosteroids (e.g., JAK inhibitor monotherapy), while higher-risk cases benefit from a more intense regimen.

Future management strategies are likely to incorporate early biomarker-guided intervention, organ-specific therapeutics, and real-time immune monitoring to adjust treatment intensity [41]. For example, patients with high MAP scores may benefit from early intensification with JAK inhibitors, while those with fibrotic features may require non-T cell-targeted therapies such as axatilimab or ROCK2 inhibitors [41]. Additionally, organ-specific therapeutics are important for personalized approaches in the management of GVHD, as different organs respond variably to treatment agents. For example, GI GVHD had a greater response to JAK inhibitors and sirolimus compared to rituximab, while rituximab demonstrated stronger efficacy for musculoskeletal GVHD [63].

Novel cellular therapies, including CAR-Tregs, NK cells, and adoptive regulatory T cell transfer, are being investigated to induce durable tolerance while preserving graft-versus-leukemia (GvL) effects [64,65,66,67]. Strategies such as fecal microbiota transplantation and combination checkpoint modulation are also under investigation to restore mucosal immunity and balance alloimmune responses [68,69,70,71,72].

## 7. Conclusions

Over the past decade, significant advances in the understanding of GVHD pathophysiology have led to the development of more precise and effective therapies. Central to this evolution is the emergence of JAK2 inhibition, with ruxolitinib now firmly established as a standard of care for steroid-refractory acute and chronic GVHD. Its success has validated the therapeutic targeting of inflammatory cytokine signaling pathways, while also revealing limitations in long-term safety and response durability.

The recent FDA approvals of axatilimab and remestemcel-L represent important milestones that expand the therapeutic toolbox and highlight the growing emphasis on non-T cell-targeted and tissue-reparative strategies. Early data on combinatorial approaches—including JAK inhibitors with macrophage-targeted agents or mesenchymal stromal cells—suggest potential for deeper immune modulation with preservation of the graft-versus-leukemia effect.

Looking ahead, biomarker-driven personalization of therapy, incorporation of organ-specific interventions, and use of novel cellular and biologic agents will likely shape the next generation of GVHD management. Continued clinical trial enrollment, real-world data collection, and mechanistic studies will be critical to refining treatment algorithms and improving long-term outcomes for transplant recipients. As the field shifts toward precision immunomodulation, integrating JAK2 inhibition into broader, individualized GVHD treatment paradigms will be central to advancing both efficacy and quality of life in allo-HSCT survivors.

## Figures and Tables

**Figure 1 biomedicines-13-01527-f001:**
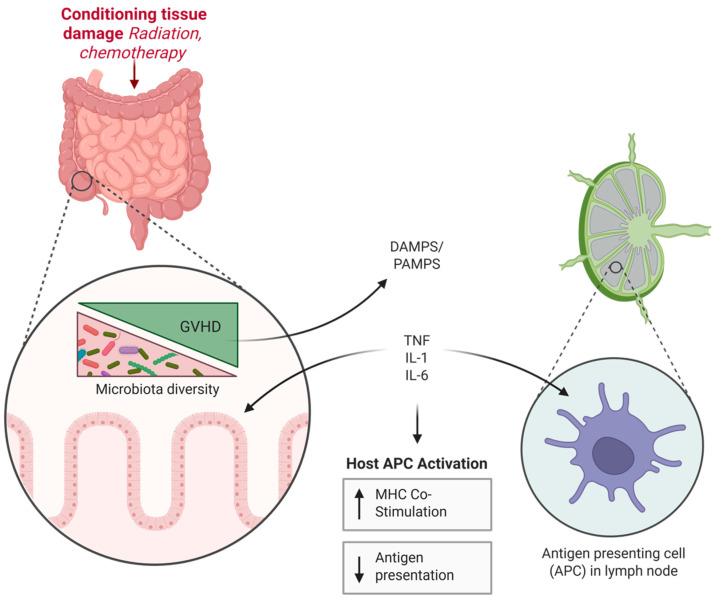
Mechanism of GVHD.

**Table 1 biomedicines-13-01527-t001:** Overview of JAK2 inhibitors and their clinical profiles.

Drug/Therapy	Target(s)	Indication	Mechanism	Key Trial/Study	FDA Approval	Phase	Route	Clinical Efficacy	Toxicity Profile
Ruxolitinib	JAK1/2 Inhibitor	Steroid-refractory aGVHD and cGVHD	Blocks JAK1/2, reduces IFN-γ, IL-6	REACH2, REACH3	Yes (2019/2021)	Approved	Oral	High ORR in both aGVHD and cGVHD; durable responses	Cytopenias (anemia, thrombocytopenia), infections
Baricitinib	JAK1/2 Inhibitor (JAK3 sparing)	Investigational GVHD therapy	Effective Treg preservation and Th1/Th17 suppression; promotes epithelial healing	Preclinical, early-phase	No	Preclinical	Oral	Effective in preclinical models; improved Treg:Teff ratio	Thrombotic risk, lipid alterations (preclinical)
Itacitinib	JAK1-selective Inhibitor	Frontline aGVHD (modest effect)	Selective JAK1 blockade	GRAVITAS-301	No	Phase III	Oral	Modest ORR; failed to meet primary endpoint	Less cytopenia; GI toxicity, infections
Pacritinib	JAK2/FLT3 Inhibitor	Investigational GVHD therapy	Suppresses STAT3^+^ CD4^+^ T cells; preserves GvL effect	Preclinical + Phase I	No	Phase I	Oral	Promising preclinical efficacy; early clinical data emerging	GI upset, cytopenias, potential QTc prolongation
Axatilimab	CSF-1R Monoclonal Antibody	Refractory cGVHD	Depletes CSF-1R–dependent profibrotic macrophages	AGAVE-201	Yes (2024)	Approved	IV	ORR 50–74% in r/r cGVHD; improvement in fibrotic symptoms	Mild transaminitis, periorbital edema
Remestemcel-L	Mesenchymal Stromal Cell Therapy	Pediatric steroid-refractory aGVHD	Promotes Tregs, suppresses Th1/Th17, secretes IL-10, TGF-β	GVHD001 (Phase III)	Yes (2024)	Approved	IV	ORR 70.4% at day 28; durable through day 100	Excellent safety; no grade ≥ 3 infusion reactions

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
