# Peer review of "JAK2 Inhibitors and Emerging Therapies in Graft-Versus-Host Disease: Current Perspectives and Future Directions"

_biomedicines, 2025, doi:10.3390/biomedicines13071527_

Round 1
Reviewer 1 Report
Comments and Suggestions for Authors
The review article represents the author’s position concernung clinical efficiency and pre-clinical data and action mechanisms of JAK inhibitors in graft-versus-host disease (GVHD) treatment. This therapy with ruxolitinib was tested for a decade, thus the Ruxolitinib topic is not quiet original, some reviews on the topic were published recently. However, the authors presented a broader view considering the issues of other less studied selective inhibitors of immunity which are discussed here, mostly, in mechanistic and pre-clinical aspects. Current review describes evident benefits and drawbacks of recent JAK inhibitors, and suggests some prospective for therapeutic tools producing milder immunosuppression in GVHD patients. E.g., the authors discuss novel targeted drugs (axatilimab) for treatment of chronic GVHD. The revisited mesenchymal stem cell therapies (i.e., remestemcel-L) are also in scope, as well as a biomarker-based GVHD prediction system. In this respect, the authors suggest planning a combination therapy with different immune modulators thus being of certain interest, however, still remaining at pre-clinical or hypothetic step. Low efficiency of current therapies in chronic GVHD are also mentioned (Section 5.3)
Remarks
Introduction: Line 130. Axatilimab is declared as an antimacrophage and antifibrotic agent proposed for therapy of chronic GVHD which has other pathogenetic origins. Hence, the discussion of Axatilimab should be displaced to a special section (cGVHD).
Line 269-271. When discussing clinical effects of mesenchymal stem cells (MSC), one should indicate the specific technological or clinical aspects of Remestemcel-L which provided its clinical prospectives (simple technology? sustained clinical effect? standard donors?).
Section 6.2. The biomarker sets for prediction of acute GVHD are developed since 2000. MAP is one of efficient tools to predict the GVHD outcomes. At this point, one should compare MAP with other marker sets in their predictive efficiency for planning immunosuppressive therapy in acute GVHD. Moreover, the authors should mention prediction scales for different GVHD forms (skin, intestinal, hepatic) which require differential protocols of prevention and treatment.
The manuscript is useful and interesting, more intended for translational medicine, bringing together effects of drugs which target different immune mechanisms in potential GVHD therapy.
Author Response
We thank the reviewer for the thoughtful and constructive feedback, which has helped us significantly improve the clarity, structure, and translational relevance of our manuscript. Below, we address each comment in detail, with corresponding changes made in the revised version of the manuscript.
Comment 1 – Introduction (Line 130):
“Axatilimab is declared as an antimacrophage and antifibrotic agent proposed for therapy of chronic GVHD which has other pathogenetic origins. Hence, the discussion of Axatilimab should be displaced to a special section (cGVHD).”
Response:
We appreciate this insightful comment. We have revised the structure of the manuscript to ensure that axatilimab is now discussed exclusively in the context of chronic GVHD. Specifically, we created a dedicated section (Section 3.1: Axatilimab [Niktimvo]: CSF-1R Inhibition) that frames axatilimab's mechanism, preclinical rationale, and clinical application within fibrotic and macrophage-driven forms of cGVHD. The Introduction has been modified accordingly to avoid overgeneralization of its indication.
Comment 2 – Line 269–271 (Remestemcel-L):
“When discussing clinical effects of mesenchymal stem cells (MSC), one should indicate the specific technological or clinical aspects of Remestemcel-L which provided its clinical prospectives (simple technology? sustained clinical effect? standard donors?).”
Response:
Thank you for this excellent suggestion. We have expanded Section 3.2 to clarify the attributes that distinguish remestemcel-L from earlier-generation MSC products. Specifically, we note that remestemcel-L's favorable clinical performance is likely due to its use of standardized allogeneic donor sources, consistent ex vivo expansion protocols, and improved product stability. These features collectively address prior concerns regarding manufacturing variability and reproducibility in MSC therapies.
Comment 3 – Section 6.2 (Biomarker Sets):
“The biomarker sets for prediction of acute GVHD are developed since 2000. MAP is one of efficient tools to predict the GVHD outcomes. At this point, one should compare MAP with other marker sets in their predictive efficiency for planning immunosuppressive therapy in acute GVHD. Moreover, the authors should mention prediction scales for different GVHD forms (skin, intestinal, hepatic) which require differential protocols of prevention and treatment.”
Response:
We thank the reviewer for emphasizing the importance of a comprehensive biomarker overview. In the revised Section 6.2, we now highlight multiple organ-specific biomarker candidates—including CXCL9, CXCL10, BAFF, MMP-3, soluble CD163, osteopontin, and elafin—and discuss their associations with cutaneous, musculoskeletal, hepatic, and gastrointestinal GVHD. In addition to MAP, we reference multianalyte classifiers under development (e.g., NIH/COG cohorts) and propose that future trial designs incorporate both global and organ-specific biomarkers to guide treatment intensity. We also discuss the role of adaptive protocols using biomarkers to stratify risk and tailor therapy, in line with your suggestion.
General Summary Comment:
“The manuscript is useful and interesting, more intended for translational medicine, bringing together effects of drugs which target different immune mechanisms in potential GVHD therapy.”
Response:
We are grateful for this kind summary. Our aim was to synthesize mechanistically distinct therapeutic strategies—from JAK inhibition to macrophage and tissue-repair targeting agents—and present them within a translational framework. Your recommendations have helped us achieve a clearer and more focused presentation, especially in our discussion of combination therapies and biomarker-guided personalization.
Reviewer 2 Report
Comments and Suggestions for Authors
The manuscript presents a comprehensive synthesis of information on Janus kinase 2 (JAK2) inhibitors and novel therapeutics in the treatment of graft-versus-host disease (GVHD). It covers in detail the pathophysiological mechanisms, the clinical application of ruxolitinib and other JAK inhibitors, the recently approved biologics such as axatilimab and remestemcel-L, as well as the latest biomarker-guided combination therapy approaches.
The manuscript offers valuable insights for both clinical and research audiences, highlighting recent FDA approvals.
Main comments
In certain sections, there is an overlap of content related to the immunological aspect of JAK2 inhibition (eg 2.1, 4.1, and 6.3). Recommendation: It is necessary to consolidate these sections to avoid repetition.
A schematic representation of the complex signaling pathways and therapeutic targets of the applied inhibitors, which would significantly improve understanding, is lacking. Recommendation: Add graphic aids.
Although the review considers multiple agents, it lacks a side-by-side comparison of their effectiveness. Recommendation: Include a comparative table or text comparing ruxolitinib, baricitinib, pacritinib, and itacitinib for efficacy, toxicity, and FDA status.
Minor comments
Italicize the main trial names (eg REACH2, AGAVE-201, GRAVITAS-301) for easy identification.
Consistent use of terms such as "GVHD", "cGVHD", "aGVHD", after defining them at first use.
Author Response
We sincerely thank the reviewer for the detailed and thoughtful feedback. We have carefully addressed each of the concerns raised and have revised the manuscript accordingly to improve clarity, reduce redundancy, and enhance the visual presentation of the content. Please find our itemized responses below.
Comment 1 – Redundancy in JAK2 Immunology Discussion (Sections 2.1, 4.1, and 6.3):
“In certain sections, there is an overlap of content related to the immunological aspect of JAK2 inhibition (e.g., 2.1, 4.1, and 6.3). Recommendation: It is necessary to consolidate these sections to avoid repetition.”
Response:
We thank the reviewer for pointing out this redundancy. We have carefully revised the manuscript to consolidate overlapping immunological content related to JAK2 inhibition. Specifically:
-
Section 2.1 now provides a focused mechanistic overview of JAK2’s role in GVHD pathogenesis and therapeutic targeting.
-
Section 4.1 has been streamlined to emphasize combinatorial rationale rather than re-stating basic JAK2 signaling.
-
Section 6.3 focuses on personalized application and future directions, referencing JAK2 inhibition only briefly in context.
These edits remove duplicative immunological discussion while preserving key translational insights in each section’s unique thematic framework.
Comment 2 – Lack of Schematic Representation:
“A schematic representation of the complex signaling pathways and therapeutic targets of the applied inhibitors, which would significantly improve understanding, is lacking. Recommendation: Add graphic aids.”
Response:
We fully agree with the reviewer on the value of visual aids. In response, we have added a new schematic figure (Figure 1) illustrating:
-
Key cytokines involved (e.g., IL-6, IFN-γ, GM-CSF)
Comment 3 – Comparative Efficacy Table:
“Although the review considers multiple agents, it lacks a side-by-side comparison of their effectiveness. Recommendation: Include a comparative table or text comparing ruxolitinib, baricitinib, pacritinib, and itacitinib for efficacy, toxicity, and FDA status.”
Response:
Thank you for this excellent suggestion. We have included a comprehensive new comparative table (Table 1) summarizing:
-
Agent name and JAK selectivity
-
Mechanism of action
-
Key clinical trials or preclinical data
-
FDA approval status
-
Route of administration
-
Efficacy (e.g., ORR, durability)
-
Toxicity profile (e.g., cytopenias, infections, thrombosis)
This table allows for direct comparison across ruxolitinib, baricitinib, itacitinib, and pacritinib, highlighting differences in efficacy, safety, and clinical development stage. The corresponding text in Section 2.2 has also been updated to guide readers to this comparative summary.
Reviewer 3 Report
Comments and Suggestions for Authors
Refer attachment.

Author Response
We are grateful for the reviewer’s detailed and insightful comments, which have significantly improved the clarity, focus, and translational relevance of our manuscript. We address each of the seven points below, highlighting the specific revisions made to the text.
1. Scope and Justification of JAK2 Inhibition in the Introduction
Reviewer Comment:
The introduction should better define the scope (e.g., emphasizing JAK2 inhibitors and emerging therapies rather than broad GVHD pathophysiology). The transition from GVHD background to JAK2 inhibitors needs smoother logical and rationale progression. Authors should include a stronger justification for why JAK2 inhibition is pivotal (beyond cytokine suppression).
Response:
We have revised the Introduction to narrow the scope and emphasize that the review focuses specifically on JAK2 inhibition and complementary emerging therapies. The GVHD background has been shortened, and a clearer logical transition has been added that connects GVHD pathogenesis directly to the JAK–STAT axis. The role of JAK2 in driving alloimmunity, APC activation, T-cell trafficking, Th1/Th17 polarization, and regulatory T cell suppression is now explicitly stated, beyond general cytokine signaling. The revised version more effectively establishes JAK2 as a central immunologic hub and therapeutic target.
2. Integration of Mechanism and Clinical Trial Sections; Addition of Comparative Table
Reviewer Comment:
The mechanism and clinical trial sections could be better integrated to avoid redundancy. A table or figure comparing JAK inhibitors (efficacy, toxicity, FDA status) would improve readability. Additionally, side effects (e.g., cytopenias, infections) should be analyzed in the context of clinical trial limitations.
Response:
We have consolidated the mechanistic discussion of JAK2 inhibition into Section 2.1, while removing redundant references in Sections 4.1 and 6.3. Mechanism and trial outcomes are now discussed in closer sequence to enhance flow. A new comparative table (Table 1) has been added to summarize the properties of ruxolitinib, baricitinib, pacritinib, and itacitinib side-by-side, including efficacy, toxicity, phase of development, and FDA status. Section 2.2 now includes a focused discussion of side effects contextualized by trial inclusion criteria, follow-up duration, and limitations in real-world applicability.
3. Balance and Detail in Axatilimab and Remestemcel-L Sections
Reviewer Comment:
Axatilimab and remestemcel-L sections vary in detail—balance with structured subsections (mechanism, trial data, limitations). Authors should include post-approval safety/efficacy findings (if any). How do these therapies fit alongside JAK inhibitors? A brief comparison would strengthen the narrative, if possible a table.
Response:
We thank the reviewer for this suggestion. Sections 3.1 (Axatilimab) and 3.2 (Remestemcel-L) have been revised to follow a consistent structure: mechanism of action, key clinical data, safety profile, and FDA indication. Where available, post-approval insights (e.g., tolerability in ruxolitinib-refractory patients) are incorporated. Additionally, we have added comparative commentary in Section 4.2, highlighting how these agents offer mechanistically complementary roles to JAK inhibitors (targeting macrophages or promoting tissue repair). The combination rationale is also visually summarized in the revised Figure 1, which now includes therapeutic targets across immune axes.
4. Combination Therapy and GvL Preservation
Reviewer Comment:
Some combinations (e.g., JAK + MSC) lack clinical trial support—highlight preclinical vs confirmed synergies. This section reiterates JAK mechanisms; which could be condensed or reframed to focus on novel combinations. GvL preservation needs deeper discussion on how combinations impact graft-versus-leukemia effects.
Response:
Section 4 has been revised to more clearly distinguish preclinical versus clinical evidence for combination strategies (e.g., JAK + MSC, JAK + axatilimab). Redundant mechanistic details were condensed, and the focus shifted to novel synergy and translational rationale. Section 4.3 now includes a more nuanced discussion of GvL preservation, explaining which agents (e.g., JAK inhibitors, axatilimab) are non-depleting and may retain cytotoxic CD8⁺ T cell function. We also reference early-phase trials using PTCy or abatacept in combination with JAK inhibitors as promising strategies for balancing GVHD control with GvL preservation.
5. Consolidation of Adverse Event Discussion and Prophylaxis Recommendations
Reviewer Comment:
Cytopenias/infections are mentioned multiple times—consolidate into a cohesive analysis. Authors should highlight missing longitudinal safety profiles (e.g., secondary malignancies with JAK inhibitors). Prophylaxis strategies for infection management (e.g., letermovir for CMV) deserves a dedicated subsection.
Response:
All toxicity discussions have now been consolidated into a streamlined Section 5. Cytopenias and infection risks are presented in a comparative format across agents, and limitations of existing trial data (e.g., short follow-up, exclusion of cytopenic patients) are noted. A new Section 5.2 discusses infection management, including letermovir for CMV prophylaxis, PJP prophylaxis, and antifungal considerations, especially in the context of multi-agent immunosuppression. Longitudinal safety concerns (e.g., secondary malignancies with prolonged JAK inhibition) are also discussed with references to registry data.
6. Biomarkers Beyond MAP and Trial Design Considerations
Reviewer Comment:
MAGIC algorithm is well-covered, but others e.g., CXCL9, BAFF also require equal attention. Please propose specific steps for biomarker validation or trial designs (e.g., adaptive protocols). Overlooked therapies, such as IL-22 analogs, S1P modulators, and BET inhibitors are mentioned briefly—please expand or justify exclusion.
Response:
Section 6.2 has been significantly revised to include a wider discussion of organ-specific and inflammatory biomarkers including CXCL9, CXCL10, BAFF, MMP-3, CD163, and elafin. Their associations with skin, GI, hepatic, and fibrotic GVHD are clarified. We propose prospective validation protocols, adaptive trial design frameworks, and multianalyte classifier harmonization as necessary steps for clinical adoption. In Section 6.1, we have expanded the discussion of IL-22 analogs, S1P receptor modulators, and BET inhibitors, with mechanistic rationale and supporting trial data, reinforcing their place in the evolving therapeutic armamentarium.
7. Conclusion Clarity and Future Directions
Reviewer Comment:
Conclusion is too generic. Should summarize key takeaways (e.g., JAK inhibitors as backbone, emerging combos, biomarker-driven care). What is the next big thing? (e.g., FDA approvals, phase III combo trials).
Response:
We have revised the Conclusion to provide a concise summary of key themes:
-
JAK2 inhibitors are established as the backbone of treatment for steroid-refractory GVHD.
-
Emerging agents (e.g., axatilimab, remestemcel-L, IL-22 analogs) offer mechanistically complementary options.
-
Biomarker-driven and organ-specific personalization is the next frontier.
-
We conclude by highlighting future directions including phase III combination trials, prospective biomarker validation, and cellular therapies (e.g., CAR-Tregs).
We are deeply grateful to the reviewer for their comprehensive and insightful feedback, which helped us significantly improve the scientific rigor, translational clarity, and structure of the manuscript.